# IDENTIFIABLE STATE DISENTANGLEMENT FOR REINFORCEMENT LEARNING WITH POLICY OPTIMALITY

## ABSTRACT

Recent advancements in reinforcement learning (RL) have showcased remarkable effectiveness in optimizing policies amidst noise. However, these endeavours have often overlooked the critical aspect of independence between the signal and noise within latent spaces, leading to performance limitations. To address this concern, we begin by conducting a thorough analysis of the identifiability of latent signal and latent noise. This analysis implicitly underscores the pivotal role of independence between these two components in RL.

A fundamental characteristic of the state variable is its role as the causal parent of the expected accumulated reward, providing a natural indicator for disentangling the state from noise. Leveraging this indicator, we demonstrate that a constrained state estimation function capable of accurately recovering both the transition and reward can precisely disentangle state and noise in scenarios characterized by deterministic dynamics with stochastic noise. In the case of stochastic dynamics, our constrained state estimation function ensures an optimal policy, even when the state and noise may not be fully disentangled. We then translate these theoretical findings into a novel methodology, which effectively isolates the signal from the noise within the latent space. This is achieved through the seamless integration of structural independence and statistical independence into a unified framework. Structurally, our proposed method employs two distinct decoders for latent signal and latent noise, enabling each decoder to capture exclusive features specific to its respective space. Statistically, the independence between latent signal and latent noise is enforced through a reward preservation constraint. Empirical evidence from extensive benchmark control tasks attests to the superiority of the proposed approach over existing algorithms in the effective disentanglement of signals from noise.

## 1 INTRODUCTION

Model-Based Reinforcement Learning (MBRL) is a Reinforcement Learning methodology that integrates a learned environment model, commonly referred to as the "world model", with a planning algorithm to guide an agent's decision-making process (Polydoros & Nalpantidis, 2017). The world model predicts future states and rewards based on interactions between the agent and the environment, while the planning algorithm utilizes these predictions to simulate potential scenarios and formulate the agent's policy. Recently, there has been a surge of interest in learning latent-space world models. These models map high-dimensional observation data, such as images, to an abstract latent representation, effectively capturing the dynamics of the environment within this learned latent space (Ha & Schmidhuber, 2018; Hafner et al., 2019; 2020; Sekar et al., 2020; Hafner et al., 2023). This approach offers the advantage of simplifying complex environments, reducing computational demands, and potentially enhancing policy generalization. However, a common assumption in these methods is that observed data is noise-free. In practical applications where noise is prevalent, this can significantly hinder the effectiveness of these techniques. This challenge arises from the difficulty of disentangling the reward-related aspects or "signals", which cannot be readily separated from the noise-related components. Such entanglement can introduce distractions during the learning process, ultimately leading to suboptimal performance (Efroni et al., 2021).

Addressing the challenge of disentangling signals from noisy observations has garnered substantial attention across various domains in machine learning, encompassing areas such as domain adapta-

tion/generalization (Von Kügelgen et al., 2021), nonlinear Independent Component Analysis (ICA) (Khemakhem et al., 2020), and others (Chen et al., 2018). These endeavours offer theoretical assurances for recovering signals from noisy observations, assuming the presence of an invertible mapping between the observation space and the latent state space.

In the realm of reinforcement learning, Huang et al. (2022) and Liu et al. (2023) have introduced akin techniques for extracting signals from noisy observations, leveraging temporal causal models. However, their reliance on the invertible mapping assumption confines their theory to fully observable Markov Decision Processes (MDPs) or very specific forms of Partially Observable Markov Decision Processes (POMDPs). In contrast, Wang et al. (2022) and Fu et al. (2021) have proposed two distinct reinforcement learning frameworks to disentangle signals from noisy observations, albeit without offering any theoretical guarantees for recovering the accurate signal.

Huang et al. (2022); Liu et al. (2023) approach the problem of disentangling signal from noise as a causal identifiability challenge, relying on observational data. Consequently, their methodology may necessitate strong assumptions regarding the geometry of the state space, as well as assumptions concerning the mapping from state/noise to observations. In the domain of reinforcement learning, however, we possess the ability to alter policies, thereby influencing the distribution of accumulated rewards. This affords us interventional data from a causal perspective, allowing for more lenient assumptions about the state space and the state/noise to observation mapping.

To be more precise, we can ascertain the state variable by evaluating whether it induces a change in expected accumulated rewards. Leveraging the Markovian nature of Markov Decision Processes (MDPs), this is equivalent to examining transition reservations and reward reservations. This unique property enables us to establish a novel method for achieving causal identifiability in the state/noise disentanglement problem within the context of reinforcement learning. Subject to certain assumptions, we demonstrate that it is possible to disentangle noise from the state in the presence of noisy observations. Moreover, when these assumptions are violated, our results indicate that the optimal policy based on our estimated state aligns exactly with the optimal policy derived from the true underlying state representation.

Drawing upon these theoretical findings, we have devised a novel approach for learning world models, enabling the disentanglement of state and noise. Our world model includes two recurrent state space models (RSSMs) to faithfully model the underlying dynamics of state and noise and thus the transition preservation and noise preservation conditions in our theory are preserved. Experimental evaluations conducted on the Deepmind Control Suite and Robodesk demonstrate that our algorithm consistently outperforms previous methodologies by a substantial margin.

## 2 PRELIMINARIES AND RELATED WORKS

### 2.1 MARKOV DECISION PROCESS

The Partial Observable Markov Decision Process (POMDP) can be defined as a 7-tuple $(\mathcal{S}, \mathcal{A}, \mathcal{O}, \mathcal{T}, \mathcal{M}, R, \gamma)$, where $\mathcal{S}, \mathcal{A}$ and $\mathcal{O}$ represent the state, action, and observation space of the problem, $\mathcal{T} = p(s'|a, s)$ denotes the probability of transition to state $s'$ given state $s \in \mathcal{S}$ and action $a \in \mathcal{A}$, $o = \mathcal{M}(s', z)$ denotes the noisy observation function where $z$ denotes the noise, and $R(s, a)$ is a real-valued reward function encoding the task objective. Finally, $\gamma$ is a discounting factor that characterizes the decay of rewards with time. The POMDP can be represented by a temporal causal model shown in Figure 1. When the state is directly observable, the POMDP is reduced to a MDP defined as a 5-tuple $(\mathcal{S}, \mathcal{A}, \mathcal{T}, R, \gamma)$.

In practice, the true underlying state is often not directly observable. Thus assume that we have a state estimation function $g : \mathcal{O} \mapsto \mathcal{S}_g$, we can always have a derived MDP $(\mathcal{S}_g, \mathcal{A}, \mathcal{T}_g, R_g, \gamma)$, where

$$\mathcal{S}_g = \{\hat{s}|\hat{s} = g(o), o \in \mathcal{O}\}, \tag{1a}$$

$$\mathcal{T}(\hat{s}', a, \hat{s}) = \sum_{o':g(o')=\hat{s}'} \sum_{o:g(o)=\hat{s}} Pr(o'|a, o), \tag{1b}$$

$$R_g(\hat{s} = g(o), a) = \mathbb{E}_{s \sim \text{Pr}(s|o)} R(s, a), \tag{1c}$$

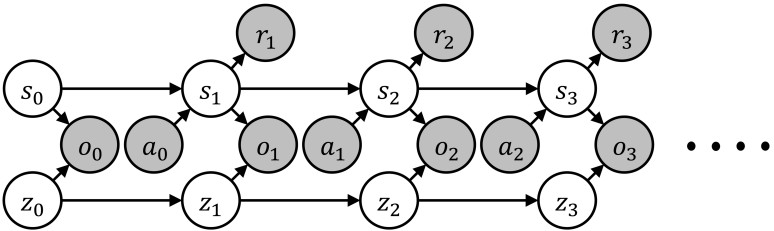

Figure 1: The temporal causal model for reinforcement learning on Markov Decision Process (MDP) with noisy observation, where $s$ denotes the latent signals while $z$ represents the latent noise, both of which remain unobservable. Each edge in the model signifies a causal direction. The nodes shaded in grey correspond to the elements of action $a$, reward $r$, and visual observation $o$ that are observable. In the context of reinforcement learning, the change of policy with change the distribution of action, and thus can be viewed as intervention. This intervention can be used to identify if part of the observation will result in a change in the expected accumulated reward and thus we are able to disentangle state and noise for a wide range of MDPs.

usually, the derived MDP will not be equivalent to the original MDP denoted by $(\mathcal{S}, \mathcal{A}, \mathcal{T}, R, \gamma)$. For General POMDP, it would not be possible to recover the state from pure observation. In this case, we will first consider a specific POMDP, where the observation function $\mathcal{M}$ is invertible. In this case, one way to ensure the equivalence is to ensure that the existence of bijection between the estimated state $\hat{s}$ and the underlying state $s$, which is also known as bisimulation, or block identifiability of the causal model (Huang et al., 2022; Liu et al., 2023). For the non-invertible observation function, we will convert the POMDP to belief-MDP where the state is a distribution over the original state space.

## 2.2 REPRESENTATION LEARNING IN RL

In recent years, Markovian representation learning in RL has employed various algorithms, including bisimulations (Gelada et al., 2019; Zhang et al., 2020), contrastive augmentation(Laskin et al., 2020; Misra et al., 2020; Deng et al., 2022) endeavour to represent states by distinguishing them in a contrastive manner. (Dann et al., 2018; Du et al., 2019; Huang et al., 2022) have proven that the minimal representation for the POMDP can be extracted from rich observations with specific constraints. The efficacy of reconstruction-based model-based reinforcement learning (MBRL) in optimizing policies by backpropagating through latent-space rollouts has been practically demonstrated in the Dreamer(Hafner et al., 2019), DreamerV2(Hafner et al., 2020), and DreamerV3(Hafner et al., 2023). On top of dreamers, denoising worlds models were proposed, including, TIA (Fu et al., 2021), Denoised-MDP (Wang et al., 2022) and (Liu et al., 2023).

## 3 LEARNING TO DISENTANGLE STATE AND NOISE

We focus on MDPs with discrete state, noise, and observation spaces, which can be interpreted as a causal model, as depicted in Figure 1. We first lay down the theoretical groundwork for the disentanglement of the signal and noise within MDPs under some mild assumptions. Subsequently, we affirm that policy optimality is indeed achieved even if the assumptions are not met. Finally, we concretize our theoretical insights through the development of a dedicated learning algorithm.

### 3.1 BLOCK IDENTIFIABILITY OF STATE AND NOISE

Previously, the identifiability of disentanglement of state and noise was established in the context of continuous state and noise spaces. This involved making assumptions about the smoothness and invertibility of the observation function, as well as certain geometric properties of the state space (Liu et al., 2023; Huang et al., 2022). As these results rely on specific properties of continuous function such as smoothness, these results can not be easily extended to discrete cases. For discrete cases, we lack properties such as smoothness and thus it is more challenging. Furthermore, the

results on discrete space often can be easily extended to continuous space as we can always convert continuous state spaces into discrete ones through quantization.

In our case, we also assume the invertibility of the observation function, and for state space, we would like to assume that there is no redundancy in the state representation. Given a state as initial state, we can then apply different series of actions, which will result in different accumulated rewards. If given two different states as initial states, the result of expected accumulated reward is the same after applying any action series, the two state can be considered as equivalent and there must be some redundancy in current state representation (Bennouna et al., 2021), and it would be natural to assume that such redundancy does not exists. Finally, we would like to firstly assume that the transition is deterministic and later we seek to loose the assumption.

Our main result on the state/noise disentanglement can be stated as follows.

**Proposition 1** (Disentanglement). *Given a POMDP* $(\mathcal{S}, \mathcal{A}, \mathcal{O}, \mathcal{T}, \mathcal{M}, \mathcal{R}, \gamma)$*, assume that*

1. ***Invertible Observation function:*** $\mathcal{M}$ *is invertible;*

2. ***No redundancy:*** *for any* $s_0 \neq s_0' \in \mathcal{S}$*, there exists an integer* $n$ *and an action series* $a_0, a_1, \ldots, a_{n-1}$ *such that*

$$\sum_{s_k, 1 \leq k \leq n} \sum_{i=0}^{n-1} \gamma^i R(s_i, a_i) \prod_{j=0}^{i-1} \mathcal{T}(s_{j+1}, a_j, s_j) \neq \sum_{s_k', 1 \leq k \leq n} \sum_{i=0}^{n-1} \gamma^i R(s_i', a_i) \prod_{j=0}^{i-1} \mathcal{T}(s_{j+1}', a_j, s_j');$$

3. ***Deterministic:*** *The transition is deterministic, and the state and action space are finite.*

*There exists an state/noise estimation function* $h(o) : \mathcal{O} \mapsto \mathcal{S} \times \mathcal{Z}$ *satisfying that*

1. ***Transition preservation:*** *For any observation* $o$ *and any action, let* $\hat{s}, \hat{z} = h(o)$*, we have* $\forall \hat{s}'$
$$Pr(\hat{s}'|a, \hat{s}) = \sum_{o':\hat{s}', \hat{z}'=h(o')} Pr(o'|a, o);$$

2. ***Reward preservation:*** $\exists \hat{R}$ *such that for any state observation pair* $(s, o)$ *with* $\hat{s}, \hat{z} = h(o)$*, for any action* $a$*,* $\hat{R}(\hat{s}, a) = R(s, a)$*;*

*and the function disentangles the state and noise,* i.e. *there exists bijection between the estimated state and the true state, as well as for the noise.*

Proposition 1 actually can be viewed as pushing redundant observation representation to non-redundant state representation. For the specific POMDP where the observation function $\mathcal{M}$ is invertible, the main issue is that the same state will be mapped into multiple different observations. Thus to obtain a compact representation of the optimal policy, we need to remove the redundant part in observation (*i.e.* noise) to get the underlying state, which is equivalent to finding a minimal representation that can preserve the transition and reward.

**Remarks on state modelling** The state and noise can be further split into controllable ones and non-controllable ones as Wang et al. (2022) and Liu et al. (2023) as the dynamics of non-controllable ones are independent with action. The further disentanglement may help to provide a more compact representation of transitions in the world model. Without such disentanglement, we are still able to obtain the optimal policy. In this case, we only disentangle the state and noise based on the fact if it affects the accumulated reward or not.

## 3.2 POLICY OPTIMALITY

Proposition 1 guarantees that we can disentangle state and noise by finding a state/noise estimation function that satisfies the transition preservation and reward preservation constraints for deterministic POMDP where there is no redundancy in state representation. If the deterministic assumption on POMDP is not satisfied, the transition preservation and reward preservation condition in Proposition 1 can still be applied to find a state representation that preserves the optimal policy.

**Proposition 2.** *Given a POMDP* $(\mathcal{S}, \mathcal{A}, \mathcal{O}, \mathcal{T}, \mathcal{M}, \mathcal{R}, \gamma)$*, assume that* $\mathcal{M}$ *is invertible, there exists an invertible state/noise estimation function* $h : \mathcal{O} \mapsto \mathcal{S}' \times \mathcal{Z}'$ *satisfies that:*

1. ***Transition preservation:*** *For any observation* $o$ *and any action, let* $\hat{s}, \hat{z} = h(o)$*, we have* $\forall \hat{s}'$

$$Pr(\hat{s}'|a, \hat{s}) = \sum_{o':\hat{s}',\hat{z}'=h(o')} Pr(o'|a, o);$$

2. ***Reward preservation:*** $\exists \hat{R}$ *such that for any state observation pair* $(s, o)$ *with* $\hat{s}, \hat{z} = h(o)$*, for any action* $a$*,* $\hat{R}(\hat{s}, a) = R(s, a)$*;*

*and the optimal policy on the MDP derived by state estimation function* $h$ *(by ignoring the estimated noise) is also the optimal policy of the original MDP.*

**General Partially Observable Case:** In various real problems, it would not be possible to have an invertible observation function. More specifically, multiple different states may be mapped into the same observation. In this case, it would not be possible to recover the original state using pure observation. Fortunately, we can convert the POMDP into an equivalent full-observable belief-MDP. For the belief-MDP, we can convert it to a POMDP with an invertible observation function by using all historical observations and actions of the original POMDP as observation. Using this conversion, our Proposition 1 and Proposition 2 can disentangle state and noise for general POMDP, or at least find a state representation that leads to the same optimal policy. However in practice due to the computational cost finding the optimal policy of POMDP remains an open and challenging problem.

### 3.3 LEARNING WORLD MODEL AND POLICY

Based on Proposition 1 and Proposition 2, we can design an algorithm that can learn to disentangle the state and noise, or learn a state representation that will lead to an optimal policy if the conditions in Proposition 1 is not satisfied. Our world model including the two Recurrent State Space Models (RSSMs, (Gregor et al., 2018)) for state and noise are modeled as Variational Auto Encoder (VAE), and one reward model. Typically a standard RSSM encompasses a representation model, a transition model and an emission model, which models the dynamics of an MDP in a variational manner.

Particularly in our world model, the RSSMs serve as the $h$ function in Proposition 1 and Proposition 2, the transition preservation constraints in the two propositions are enforced by fitting the data transition with the world model transition, and the reward preservation constraints are enforced by minimizing the negative log-likelihood of estimated reward conditional on estimated state. Our notations are summarized in Eq. 2.

| Components | State-RSSM & Reward Model | Noise-RSSM | |
|---|---|---|---|
| Transition Model | $p_\psi(s_t|s_{t-1}, a_{t-1})$ | $p_\psi(z_t|z_{t-1})$ | |
| Representation Model | $q_\psi(s_t|s_{t-1}, a_{t-1}, o_t, z_t)$ | $q_\psi(z_t|z_{t-1}, o_t, s_{t-1}, a_{t-1})$ | (2) |
| Emission Model | $p_\theta(o_t^s|s_t)$ | $p_\theta(o_t^z|z_t)$ | |
| Reward Model | $\log p_\theta(r_t|s_t)$ | | |

Our priors on state and noise, denoted as $p_\psi(s_t|s_{t-1}, a_{t-1})$ and $p_\psi(z_t|z_{t-1})$ respectively, are established through two separate transition models—one for state and one for noise. Specifically, the prior for the current state (e.g., $s_t$) is derived from the transition of the previous state and action, namely, $s_{t-1}$ and $a_{t-1}$, while the prior for the current noise (e.g., $z_t$) is modelled as the transition from the previous noise, $z_{t-1}$. Given these priors, however, computing the true posterior distribution, represented as $p_\psi(s_t, z_t|s_{t-1}, a_{t-1}, z_{t-1}, o_t)$, is still generally an intractable task due to the complexity introduced by multiple integrals in the marginal distribution. To address this challenge, we employ a structural variational posterior distribution, known as the representation model within the context of RL, to provide an approximation of the true posterior distribution:

$$q_\psi(s_t, z_t|s_{t-1}, a_{t-1}, z_{t-1}, o_t) = q_\psi(s_t|s_{t-1}, a_{t-1}, o_t, z_t)q_\psi(z_t|z_{t-1}, o_t, s_{t-1}, a_{t-1}). \quad (3)$$

Note that prior research efforts (Fu et al., 2021) have sought to alleviate computational complexities by adopting a simplified factorized variational posterior. In these approaches, the factorized

variational posteriors for $s_t$ and $z_t$ are assumed to be mutually independent, conditioned on the observational data $o_t$. While this simplifies the optimization process, it does, however, deviate from the fundamental principle that $s_t$ and $z_t$ should exhibit mutual dependence, given the presence of observational data $o_t$. Driven by this consideration, we emphasize the superiority of the proposed structural variational posterior, as defined in Eq. 3. This formulation effectively captures the interdependency between $s_t$ and $z_t$ given the observational data $o_t$ offering a more accurate representation compared to the commonly used factorized variational posterior. Consequently, when combining the priors with the structural variational posterior, we derive the evidence lower bound (ELBO):

$$\mathcal{L}_{\text{ELBO}} = -\mathbb{E}\bigg[\log p(o_t|s_t, z_t) - \alpha D_{KL}(q_\psi(s_t|s_{t-1}, a_{t-1}, o_t, z_t)\|p_\psi(s_t|s_{t-1}, a_{t-1}))$$
$$- \beta D_{KL}(q_\psi(z_t|z_{t-1}, o_t, s_{t-1}, a_{t-1})\|p_\psi(z_t|z_{t-1}))) \bigg]. \tag{4}$$

In our empirical approach, we introduce two hyper-parameters, denoted as $\alpha$ and $\beta$, to effectively enforce the Kullback–Leibler divergence between the variational posterior and the prior. It is worth noting that even with the incorporation of $\alpha$ and $\beta$, the modified ELBO continues to serve as a lower bound on the marginal log-likelihood of the observational data $o_t$. Furthermore, recognizing that noise and state independently contribute to the observational data $o_t$, we employ two distinct networks to reconstruct the observational data $o_t$, expressed as $p(o_t|s_t, z_t) = p(o_t^z|z_t)p(o_t^s|s_t)$. By combining this reconstruction process with the negative log-likelihood of reward estimation, we derive the final objective as follows:

$$\mathcal{L}_{\text{obj}} = \mathcal{L}_{\text{ELBO}} - \underbrace{\mathbb{E}[\log p_\theta(r_t|s_t)]}_{\mathcal{L}_{\text{r}}}. \tag{5}$$

Overall, the proposed RSSM learning objective comprises two key components: the ELBO loss, denoted as $\mathcal{L}_{\text{ELBO}}$, and the reward loss, denoted as $\mathcal{L}_{\text{r}}$. The $\mathcal{L}_{\text{ELBO}}$ component ensures the preservation of all information contained within the observational data, which includes a mixture of state and noise information. In tandem with this, the $\mathcal{L}_{\text{r}}$ component serves the crucial purpose of disentangling state information from noise information within the mixture of state and noise information. This is particularly important since rewards depend solely on state information and are entirely independent of noise.

**Transition and Reward Preservation Constraints**   In the temporal causal model (Shown in Figure 1, the state is the only parent of In our implementation, we enforce the reward preservation constraints by maximising the log-likelihood between the estimated state and reward. Meanwhile Fu et al. (2021) attempts to minimize the dependency between noise and noise by minimising the log-likelihood between estimated state and reward, which requires a regressor that will predict a reward dislikes the real one. Thus, it in contrast will result in the estimated noise to contain information about the reward so that the regressor can give a dissimilar output. Liu et al. (2023) minimizes the dependency between reward and noise by minimizing the mutual information between them. However, the estimation of mutual information is known to be highly non-trivial, and our Proposition 2 actually implies that the independency between noise and reward can be enforced by transition preservation and reward preservation condition.

Similar to the previous work(Hafner et al., 2020; Wang et al., 2022), we also employ online learning to optimize policies through the training of an actor-critic model using latent signal trajectories. These trajectories are exclusively composed of the generated state derived from the state dynamics model since the absence of noise could improve the sample efficiency of the actor-critic model.

## 4   EXPERIMENTS

The present study aimed to assess the efficacy of a proposed method in various image observation environments, including the DeepMind Control Suite (DMC) Tunyasuvunakool et al. (2020) and RoboDesk Kannan et al. (2021). In DMC, six control tasks, where the observation space is $64 \times 64 \times 3$, and the proposed method was tested in environments with additional noisy distractors. To introduce noise, the pure blue background is replaced with a grey-scaled video sampled from Kinetics 400 "Driving car". Previous works, including TIA Fu et al. (2021), Denoisedmdp Wang et al.

(2022), and IFactor Liu et al. (2023), only chose a piece of random video as the background during the simulated episodes and data collection, which makes model learn the true dynamics easily. However, in reality, the noise is much more diverse, which makes previous methods less practical. Therefore, instead of always using the same video, The background of each simulated episode will be randomly selected from Kinetics 400. Moreover, the agent is tested by using another bunch of unseen videos. Therefore the comparison is made on 3 different versions of DMC environments, which are Noiseless DMC (original DMC), DMC with uniform background, and DMC with diverse background. RoboDesk task, where the observation space is $96 \times 96 \times 3$, is a modification from Denoisedmdp, where the agent can switch the TV to green by pushing the button and is rewarded by the greenness of the TV Wang (2022). Additional noisy distractors such as videos from Kinetics 400(the same video selected setting as that in DMC), environment flickering, and camera jittering are also included. Similarly, the training and test videos were separated.

## 4.1 BASELINES

The proposed method was compared with three model-based baselines: DreamerV3 Hafner et al. (2023), Task Informed Abstractions (TIA) Fu et al. (2021), DenoisedMDP Wang et al. (2022), and IFactorLiu et al. (2023). **DreamerV3** Hafner et al. (2023) represents the latest iteration of the Dreamer model, featuring a larger and more robust architecture optimized for general RL tasks. **TIA** Fu et al. (2021) also designed separate but symmetric dynamics and decoder for state and noise, however, the ground-truth reward when only considering latent noise as input, may not lead to the statistical independence of state and noise. **DenoisedMDP** Wang et al. (2022) . It seeks to achieve this disentanglement by separating the uncontrollable factors, controllable factors, and reward-irrelevant factors in an observation into three distinct latent variables. It utilizes a single decoder to map both latent signals and noise to reconstruct observations. **IFactor** Liu et al. (2023) is the state-of-the-art reconstruction-based MBRL method, similar to DenoisedMDP, controllable factors, and reward-irrelevant factors are taken into consideration and the inter-causal relations are identified based on the strong assumptions. TIA, DenoisedMDP and IFactor were designed for noisy environments, but the DreamerV3 is not. All the methods were trained for 1 million environment steps and evaluated every 5,000 environment steps based on the average return from 10 episodes consisting of 1,000 steps each. The evaluation metrics' mean and standard deviation are derived from three independent runs. Each DMC task takes 8 hours on a GTX 3090 GPU, while the RoboDesk task takes 15 hours. We follow the same hyper-parameters ($\alpha, \beta$, and the dimension of latent states) from DenoiedMDP without further tuning.

## 4.2 PERFORMACE ON DMC TASKS

Considering the relative ease of implementing the noiseless DMC and Uniform DMC compared to the more complex Diverse DMC, our methods may not exhibit exceptional performance in comparison with other approaches, as demonstrated in the supplementary materials (see C.2). The performance and reconstruction results are presented in Table 4.2, Figure 2, and the figures in 5. These findings demonstrate that our method not only ensures optimal disentanglement and convergence but also achieves superior performance and reconstruction outcomes. If the robot's body occupies a significant portion of the image, it will be easily identified; otherwise, its identification becomes challenging, which can be quantified by signal-to-noise ratio (SNR). Tasks with high SNR include Walker Run, Cheetah Run, and Finger Spin tasks, where disentangling state-noise relationships could potentially be facilitated. The state reconstruction results demonstrate that the compared method still encounters challenges in preserving the integrity of the state, as well as distinguishing between state and noise. For instance, in tasks such as Walker Run and Cheetah Run, DenoisedMDP and IFator inadvertently disclose certain state details to the noise. Although TIA also utilizes the separate RSSM, it still encounters challenges in distinguishing between state and noise in Finger Spin, potentially due to an incorrect learning objective. Another notable distinction between TIA and our approach lies in the utilization of asymmetric decoders for state and noise, enhancing the efficiency of disentanglement as demonstrated by our ablation study. The situation worsens with lower SNR, making other methods ineffective in identifying the robot in Ball in Cup Catch and Cartpole Swingup tasks. In contrast, our method not only separates the state from noise effectively in tasks with higher SNR but also accurately captures states even in tasks with lower SNR. For example, both the cup and ball can be captured in the task Ball in Cup Catch, the cart is captured in the Cartpole Swingup, and the Red destination is captured in Reacher Easy. The superior reconstruction state

consistently leads to enhanced performance, as demonstrated in Table 4.2. Our method only falls short in the Reacher Easy task, while DreamerV3 surpasses other denoising methods specifically designed for this purpose.

| Task | Dreamer | TIA | DenoisedMDP | IFactor | Ours |
|---|---|---|---|---|---|
| Cartpole Swingup | 167.15 ± 54.00 | 122.28 ± 13.34 | 132.97 ± 5.44 | 144.72 ± 27.86 | **195.13±19.00** |
| Cheetah Run | 144.03 ±59.20 | 351.91 ± 127.79 | 221.83 ± 41.74 | 248.35± 104.87 | **520.57 ± 130.84** |
| Ball in cup Catch | 64.77 ± 112.19 | 31.31 ± 41.27 | 50.14 ± 31.08 | 134.60 ±60.60 | **272.60 ± 136.25** |
| Finger Spin | 310.66 ± 121.82 | 282.74 ± 230.41 | 75.11 ± 64.41 | 566.38 ±344.57 | **615.01± 211.31** |
| Reacher Easy | **441.33 ± 472.06** | 86.98 ± 53.85 | 101.28 ± 19.60 | 176.55 ±83.70 | 71.43 ± 13.29 |
| Walker Run | 108.52 ±98.38 | 315.87 ± 127.00 | 74.36 ± 19.57 | 68.77 ± 20.00 | **437.20 ± 71.80** |

Table 1: The experiments were conducted on the Diverse DMC tasks, and the performances were evaluated based on the mean episode return and standard deviation from three independent runs. The best results are highlighted in **bold**. Our method outperforms others in the majority of tasks.

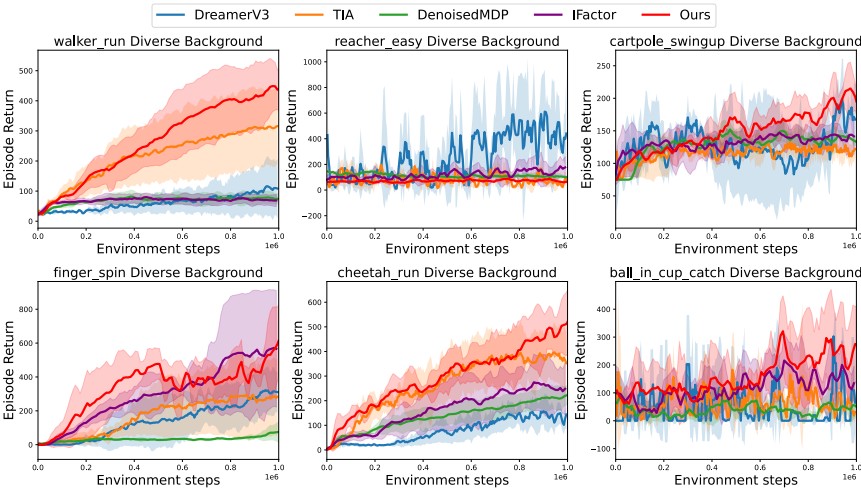

Figure 2: In the diverse video background scenario, Our method outperformed other's strong base-lines to achieve the best performance in 5 out of 6 tasks, except for the Reacher easy.

## 4.3 PERFORMANCE ON ROBODESK

Given the provision of Robodesk, which offers a comprehensive reward-related state observation (with a high signal-to-noise ratio), such as assessing the greenness of the TV screen, we find that the performance of the compared methods is closely matched. The corresponding results are presented in Figure 7.

## 4.4 ABLATION STRUDY

In the ablation study, we conducted a performance comparison between symmetric and asymmetric structures on the DMC with diverse backgrounds. The symmetric structure employed two 8-layer decoders, while the asymmetric structure consisted of an 8-layer state decoder and a 4-layer noise decoder. Table 2 and Figure 3 illustrate the disentanglement achieved by the asymmetric structures. This is because our algorithm attempts to compress all reward-related information into the compact state representation, a decoder with stronger representation power may allow better compression of the encoder. Especially, with symmetric structures, it is possible to make the model mistake the noise as the state.

| Task | Cartpole Swingup | Cheetah Run | Ball in cup Catch | Finger Spin | Reacher Easy | Walker Run |
|---|---|---|---|---|---|---|
| **Asymetric decoders** | **195.13 ± 19.00** | **520.57 ± 130.84** | **272.60 ± 136.25** | **615.01 ± 211.31** | 1.43 ± 13.29 | **437.20 ± 71.80** |
| **Symetric decoders** | 141.94 ± 10.85 | 292.49 ± 94.19 | 126.59 ± 35.30 | 269.87 ± 218.16 | **90.41 ± 22.43** | 320.56 ± 247.30 |

Table 2: This comparison is conducted on the Diverse DMC tasks, The Asymmetric structure performs better than in almost all the tasks except for the Reacher Easy. The best results are highlighted in **bold**.

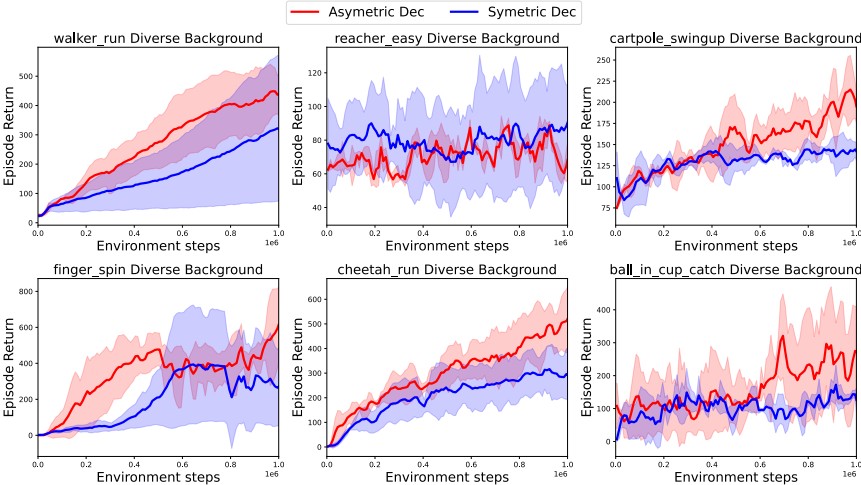

Figure 3: The Asymmetric world model demonstrates superior performance over the symmetric one in the majority of tasks within a diverse video background scenario. A high standard deviation indicates that the symmetric world model lacks stability across different random seeds.

## 5 CONCLUSION

Recent studies that delve into the intersection of causality and reinforcement learning have made notable strides. However, they often overlook certain crucial aspects of problem settings within the realm of RL. For instance, some of these works make strong assumptions about the geometry of the state space, as well as assumptions concerning the mapping from state or noise to observation. In our approach, we take a starting point from the perspective of RL, thoroughly examining these specific problem settings within the RL context. Subsequently, we demonstrate how we can substantially relax the assumptions made in previous work regarding the identification of latent state and noise. This relaxation of assumptions serves to bridge the gap between fundamental theoretical concepts and their practical applicability in RL, contributing to a more holistic understanding and application of these ideas. Furthermore, we translate our insights into a novel methodology, incorporating unique design elements such as employing two decoder network structures and enforcing independence between state and noise through KL divergence penalties. Our empirical findings, derived from comprehensive benchmark control tasks, substantiate the superiority of our proposed approach compared to existing algorithms in effectively disentangling signals from noise.

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

# A    PROOF OF PROPOSITIONS

**Proposition 1** (Disentanglement). *Given a POMDP $(\mathcal{S}, \mathcal{A}, \mathcal{O}, \mathcal{T}, \mathcal{M}, \mathcal{R}, \gamma)$, assume that*

1. ***Invertible Observation function:*** *$\mathcal{M}$ is invertible;*

2. ***No redundancy:*** *for any $s_0 \neq s_0' \in \mathcal{S}$, there exists an integer $n$ and an action series $a_0, a_1, \ldots, a_{n-1}$ such that*

$$\sum_{s_k, 1 \leq k \leq n} \sum_{i=0}^{n-1} \gamma^i R(s_i, a_i) \prod_{j=0}^{i-1} \mathcal{T}(s_{j+1}, a_j, s_j) \neq \sum_{s_k', 1 \leq k \leq n} \sum_{i=0}^{n-1} \gamma^i R(s_i', a_i) \prod_{j=0}^{i-1} \mathcal{T}(s_{j+1}', a_j, s_j');$$

3. ***Deterministic:*** *The transition is deterministic, and the state and action space are finite.*

*there exists an state/noise estimation function $h(o) : \mathcal{O} \mapsto \mathcal{S} \times \mathcal{Z}$ satisfies that*

1. ***Transition preservation:*** *For any observation $o$ and any action, let $\hat{s}, \hat{z} = h(o)$, we have $\forall \hat{s}'$*

$$Pr(\hat{s}'|a, \hat{s}) = \sum_{o':\hat{s}', \hat{z}'=h(o')} Pr(o'|a, o);$$

2. ***Reward preservation:*** *$\exists \hat{R}$ such that for any state observation pair $(s, o)$ with $\hat{s}, \hat{z} = h(o)$, for any action $a$, $\hat{R}(\hat{s}, a) = R(s, a)$;*

*and the function disentangles the state and noise,* i.e. *there exists a bijection between the estimated state and the true state, as well as for the noise.*

*Proof.* The existence of $h$ is trivial as the invert of $\mathcal{M}$ must satisfy the transition preservation and reward preservation condition. Then for a $h$ function satisfies the two conditions, it would be straightforward to show that the estimated state $\hat{s}$ is a valid representation (defined in Definition Definition 3.1 of Bennouna et al. (2021)) of the original MDP. By Proposition 3.3 of Bennouna et al. (2021), $\mathcal{S}$ is the unique minimal state representation of the MDP given the no equivalent state and deterministic assumption. Then by the fact the space of $\hat{s}$ is also $\mathcal{S}$, it would be trivial that there must exist a bijection between the estimated state and the true state. Then by the fact that $\mathcal{M}$ and $h$ are invertible, there must exist a bijection between the estimated noise and the true noise.

□

**Proposition 2.** *Given a POMDP $(\mathcal{S}, \mathcal{A}, \mathcal{O}, \mathcal{T}, \mathcal{M}, \mathcal{R}, \gamma)$, assume that $\mathcal{M}$ is invertible, there exists an invertible state/noise estimation function $h : \mathcal{O} \mapsto \mathcal{S}' \times \mathcal{Z}'$ satisfies that:*

1. ***Transition preservation:*** *For any observation $o$ and any action, let $\hat{s}, \hat{z} = h(o)$, we have $\forall \hat{s}'$*

$$Pr(\hat{s}'|a, \hat{s}) = \sum_{o':\hat{s}', \hat{z}'=h(o')} Pr(o'|a, o);$$

2. ***Reward preservation:*** *$\exists \hat{R}$ such that for any state observation pair $(s, o)$ with $\hat{s}, \hat{z} = h(o)$, for any action $a$, $\hat{R}(\hat{s}, a) = R(s, a)$;*

*and the optimal policy on the MDP derived by state estimation function $h$ (by ignoring the estimated noise) is also the optimal policy of the original MDP.*

*Proof.* The existence of $h$ is trivial as the invert of $\mathcal{M}$ must satisfy the transition preservation and reward preservation condition.

Then for a $h$ function satisfies the two conditions, we prove it leads to the optimal policy. For the invertible observation function $\mathcal{M}$, it is obvious we can use observation as a state. Thus we only need to prove that the optimal policy obtained from the estimated state is the same as from observation. Let's consider the first step of value iteration, where all initial value is set to zero.

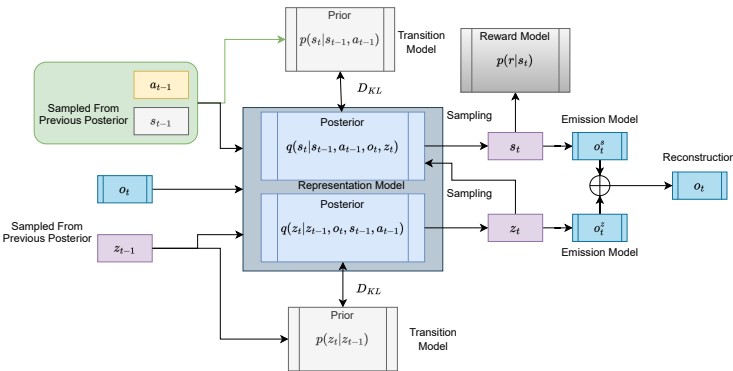

Figure 4: The overall illustration of our world model.

For all state-observation pair $(s, o)$, let $\hat{s}, \hat{z} = h(o)$, it is straightforward that the value iteration on estimated state space and observation space satisfies[1]:

$$
\begin{aligned}
Q^1(\hat{s}', a) =& \hat{R}(\hat{s}, a) + \gamma \sum_{\hat{s}'} Pr(\hat{s}'|a, \hat{s}) V^0(\hat{s}') \\
=& R(o, a) + \gamma \sum_{\hat{s}'} \sum_{o':\hat{s}', \hat{z}'=h(o')} Pr(o'|a, o) V^0(o') \\
=& R(o, a) + \gamma \sum_{o'} Pr(o'|a, o) V^0(o') = Q^1(o, a),
\end{aligned}
\tag{6a}
$$

$$
V^1(\hat{s}') = V^1(o) = \max_a Q^1(\hat{s}', a) = \max_a Q^1(o, a).
\tag{6b}
$$

Assume that after $k$ iterations of value iteration, we still have that

$$
Q^k(\hat{s}', a) = Q^k(o, a), \quad V^k(\hat{s}') = V^k(o).
\tag{7}
$$

It would be straightforward after the $k + 1$'s iteration we have

$$
\begin{aligned}
Q^{k+1}(\hat{s}', a) =& \hat{R}(\hat{s}, a) + \gamma \sum_{\hat{s}'} Pr(\hat{s}'|a, \hat{s}) V^k(\hat{s}') \\
=& R(o, a) + \gamma \sum_{\hat{s}'} \sum_{o':\hat{s}', \hat{z}'=h(o')} Pr(o'|a, o) V^k(o') \\
=& R(o, a) + \gamma \sum_{o'} Pr(o'|a, o) V^k(o') = Q^k(o, a),
\end{aligned}
\tag{8a}
$$

$$
V^{k+1}(\hat{s}') = V^{k+1}(o) = \max_a Q^{k+1}(\hat{s}', a) = \max_a Q^{k+1}(o, a).
\tag{8b}
$$

Then by mathematical induction, the value iteration procedure on the estimated state space and the original state space must match. Finally, by the fact that value iteration must converge to the optimal policy, it would be obvious that the optimal policy on the estimated state space matches the optimal policy on the original MDP. □

## B  OVERALL STRUCTURE

## C  PERFOMANCE ON DMC VATIANTS

### C.1  NOISELESS DMC

The performance of the compared method in noiseless DMC environments is illustrated in Table 3 and Chat 5. In these noiseless environments, symmetric decoders are employed for both state and

---

[1] Here we slightly abuse the notation $R(o, a) = R(s, a)$ where observation $o$ corresponds state $s$.

noise. Due to the minimal presence of noise, the raw observation can be considered a state observation, rendering only the state decoder necessary. If we were to retain only one decoder, the proposed structure would be reduced to a typical dreamer model.

| Task | Dreamer | TIA | DenoisedMDP | IFactor | Ours |
|---|---|---|---|---|---|
| **Cartpole Swingup** | $824.62 \pm 59.36$ | $716.55 \pm 151.01$ | $555.49 \pm 416.89$ | $\mathbf{836.21 \pm 26.45}$ | $824.27 \pm 12.01$ |
| **Cheetah Run** | $780.62 \pm 106.69$ | $573.56 \pm 361.39$ | $\mathbf{815.78 \pm 72.20}$ | $717.14 \pm 213.07$ | $797.16 \pm 24.96$ |
| **Ball in cup Catch** | $\mathbf{970.44 \pm 9.04}$ | $959.00 \pm 6.38$ | $956.48 \pm 8.21$ | $908.54 \pm 102.09$ | $947.77 \pm 20.35$ |
| **Finger Spin** | $\mathbf{939.22 \pm 22.22}$ | $252.66 \pm 219.10$ | $442.76 \pm 152.80$ | $695.97 \pm 248.07$ | $469.78 \pm 228.85$ |
| **Reacher Easy** | $801.44 \pm 166.39$ | $891.66 \pm 120.26$ | $680.30 \pm 259.33$ | $\mathbf{948.92 \pm 39.07}$ | $541.24 \pm 390.41$ |
| **Walker Run** | $\mathbf{762.29 \pm 103.05}$ | $553.41 \pm 55.30$ | $629.54 \pm 57.89$ | $405.77 \pm 78.61$ | $642.19 \pm 39.93$ |

Table 3: The performances of the compared methods in the noiseless DMC environments.

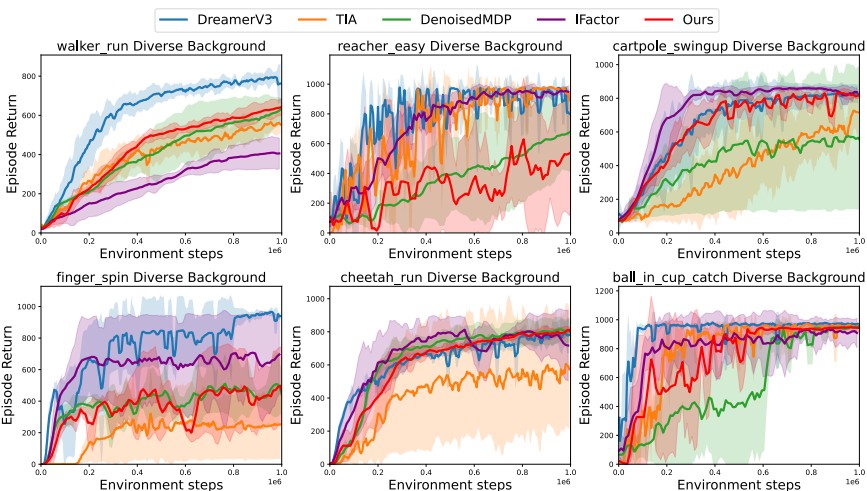

Figure 5: In the noiseless DMC, armed by a much deeper world model, DreamerV3 always can achieve the best performance.

## C.2 DMC WITH UNIFORM BACKGROUND

Previous work(TIA, DenoisedMDP and IFactor) were trained and tested their method on the Uniformed background DMCs, where the disentanglement could be easier. The performance of the compared methods is shown in Table 4 and Figure 6. Since DreamerV3 is not designed for the purpose of denoising, it is not included in the comparison on the Uniform DMC.

| Task | TIA | DenoisedMDP | IFactor | Ours |
|---|---|---|---|---|
| **Cartpole Swingup** | $119.94 \pm 16.41$ | $97.19 \pm 18.58$ | $\mathbf{209.80 \pm 354.54}$ | $172.28 \pm 70.26$ |
| **Cheetah Run** | $291.08 \pm 74.14$ | $317.31 \pm 13.66$ | $\mathbf{514.51 \pm 165.56}$ | $469.47 \pm 117.29$ |
| **Ball in cup Catch** | $52.36 \pm 49.66$ | $120.22 \pm 25.17$ | $5.63 \pm 9.75$ | $\mathbf{201.81 \pm 156.10}$ |
| **Finger Spin** | $354.44 \pm 299.01$ | $559.95 \pm 47.03$ | $\mathbf{504.3 \pm 169.35}$ | $371.04 \pm 179.50$ |
| **Reacher Easy** | $366.19 \pm 129.51$ | $639.53 \pm 118.87$ | $\mathbf{832.36 \pm 79.60}$ | $345.44 \pm 471.07$ |
| **Walker Run** | $325.18 \pm 42.01$ | $401.35 \pm 59.74$ | $212.36 \pm 166.29$ | $\mathbf{507.25 \pm 139.84}$ |

Table 4: The performances of the compared methods in Uniform DMC environments with unseen video backgrounds. The best results are in **Bold**.

## C.3 DMC WITH DIVERSE BACKGROUND

Chart 2 and Figures in 5 show the episode returns and reconstruction results for all the compared methods.

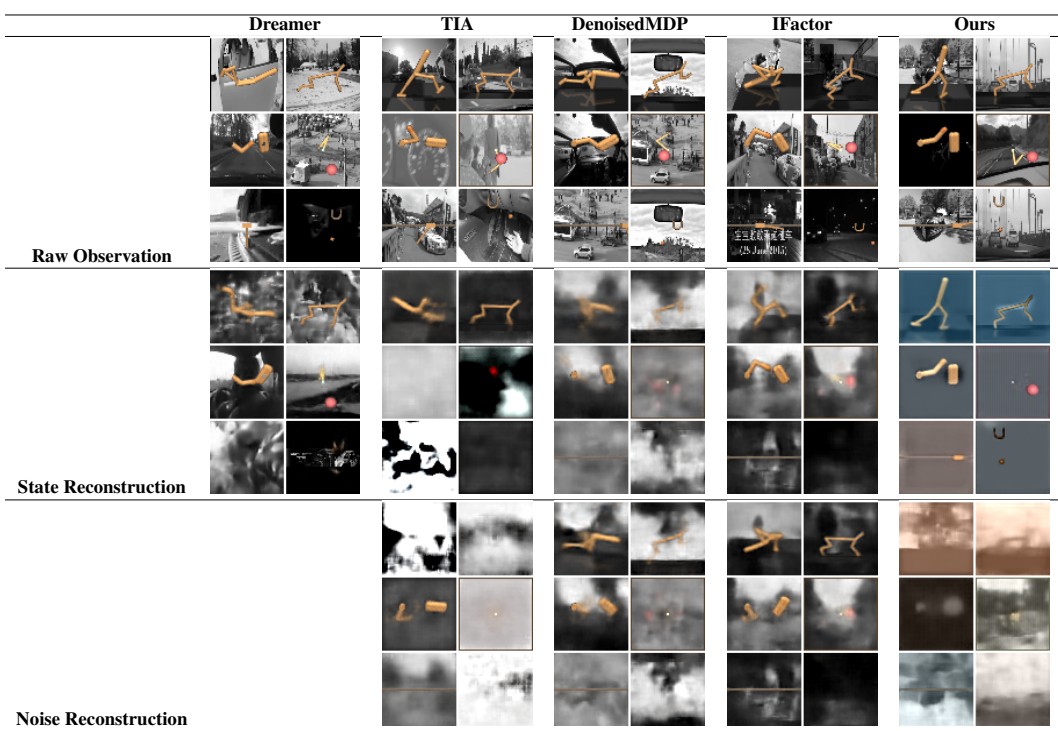

Table 5: The Reconstruction visualization of six DMC tasks, namely Walker Run, Cheetah Run, Finger Spin, Reacher Easy, Cartpole Spingup and Ball in Cup Catch, demonstrates the reconstruction of signal and noise from different world models. Our models exhibit clear decoupling between state and noise, whereas other compared methods produce numerous erroneous reconstructions and consistently sacrifice the state observation.

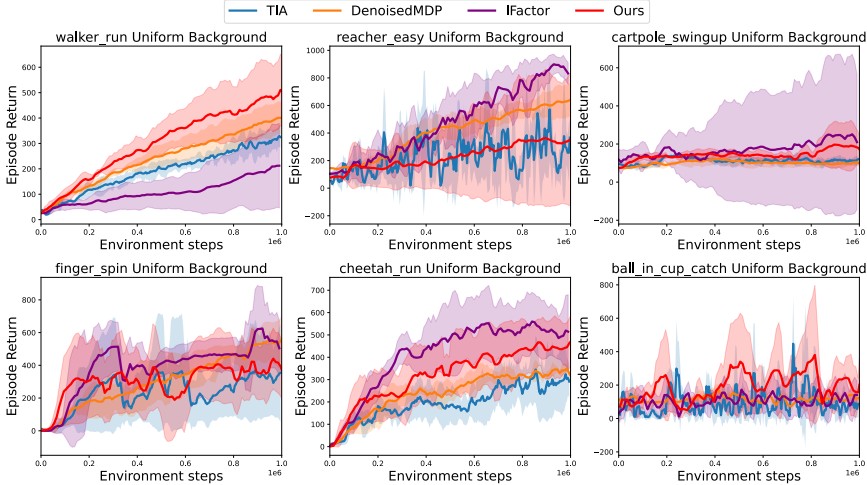

Figure 6: In the noiseless DMC, armed by a much deeper world model, DreamerV3 always can achieve the best performance.

# D    PERFORMANCE ON ROBODESK

In the Robodesk tasks, we can see that all the compared methods achieve quite close performance, around 500. However, because of the rich reward-related state, the error tolerance is much higher than DMC. Therefore, Robodesk might not be an appropriate evaluation environment.

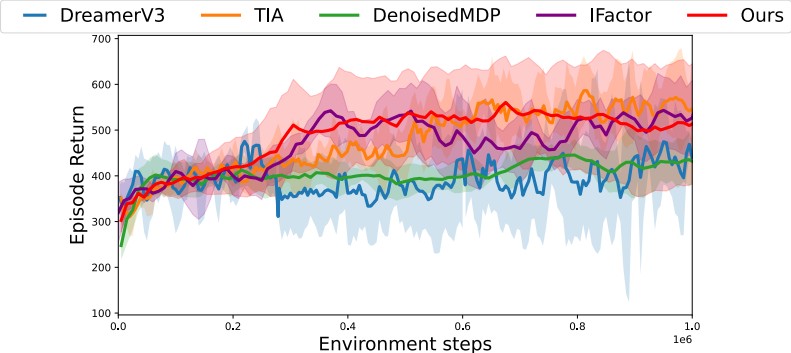

Figure 7: All compared methods demonstrate comparable performance on RoboDesk.

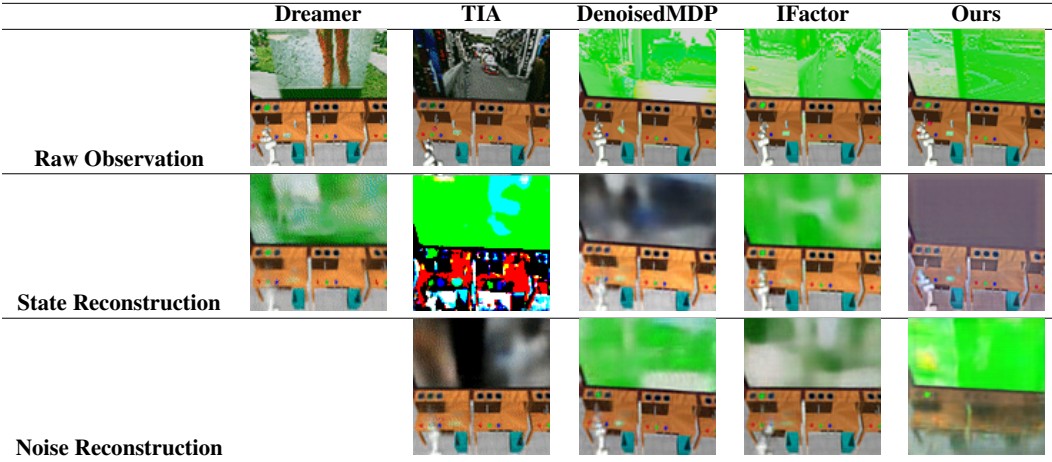

Table 6: In the Robodesk task, all methods exhibit comparable performance, but their denoising capabilities vary. TIA fails to identify the Robot arm; DenoisedMDP and our method mistake the greenness as noise; IFactor also mistakes the green light as noise. Although methods confuse the state and noise, they still get quite good performance maybe because the reward-related state observation is rich.

