# OpenReview forum: "Identifiable State Disentanglement for Reinforcement Learning with Policy Optimality"
_ICLR.cc/2024/Conference — ICLR 2024 Conference Withdrawn Submission_

### Official Review · Reviewer_vGCi · 2023-10-25

**Soundness:** 1 poor
**Presentation:** 2 fair
**Contribution:** 2 fair
**Rating:** 3
**Confidence:** 3

**Summary:**

This paper proposes a model-based reinforcement learning method to disentangle the noise and state from the observation space without breaking policy optimality.

**Strengths:**

+ This paper provides some theoretical analyses and empirical results for their method.
+ The method in this paper outperforms some advanced baselines in some tasks.

**Weaknesses:**

+ This paper is not reading-friendly. The manuscript could benefit significantly from further editing for clarity, especially the specific difference from prior works and the contribution.
+ The theory in this paper is difficult to understand because there are contradictions.
  + For instance, in the 'No redundancy' item of Proposition 1, the definition of the condition and the sum of $s_k$ is confused and not explained clearly.
  + The existence of the function $h$ is suspicious, especially after reading the proof in the Appendix. As said by this paper, the same state $s$ should be mapped into multiple different observations $o$, meanwhile $h$ is invertible. It means that $h^{-1}$ is a non-injective function that maps one input to multiple outputs. If you mean $o=h^{-1}(s,z)$ and $o$ changes with $z$, then the  'Transition preservation' property does not hold because the transition from $\hat{s}$ to $\hat{s'}$ should be calculated by both  $\sum_{o':\hat{s}',\hat{z}'=h(o')}$ and $\sum_{o:\hat{s},\hat{z}=h(o)}$. Your assumption corresponds to an extremely special case that may not exist in practice.
  + Due to your unclarity in Proposition 1, the proof of Proposition 2 is also confusing. I guess the 'Reward preservation' property should be $\hat{R}(\hat{s},a)=R(o,a)$, right? Because it is used in Eq. (6a) in the Appendix. However, how can $\sum_{\hat{s}'}P(\hat{s}'|\hat{s},a)V(\hat{s}')=\sum_{\hat{s}'}\sum_{o'}P(o'|o,a)V(o')$ hold if one $\hat{s}$ should be mapped to multiple $o$? I think there are some issues in the proof that should be checked and corrected.
+ The proposed method seems not very novel.
  + Firstly, your components in Eq. (2) are not consistent with the structure illustrated in Figure 1. In Eq. (2), $z_t$ relies on $z_{t-1}$ while $s_t$ relies on $z_t$, which is different from the contents in Figure 1. This can cause great misunderstanding.
  + Secondly, many components in the proposed method seem to be the same as the work of Liu et al. (2023).
  + Thirdly, your learning method uses the expression $p(o|s,z)=p(o^s|s)p(o^z|z)$ above Eq. (5). It means that the observation $o$ can be naturally decoupled into independent terms $o^s$ and $o^z$. If that holds, what is the usage of the representation $s$ and $z$? It may conflict with your assumption and motivation.
+ There are many works involved with states with noise in the RL domain, for instance, building Blocked-MDP to make casual decisions. Compared with previous works, this work seems to lack technical novelty.

**Questions:**

+ It seems that this work requires stronger and more special conditions than the previous related works. Can you highlight your advantages and contributions more clearly?
+ Your theoretical analysis is confusing. Please explain the questions mentioned above.

---

### Official Review · Reviewer_1xFa · 2023-10-30

**Soundness:** 1 poor
**Presentation:** 1 poor
**Contribution:** 1 poor
**Rating:** 3
**Confidence:** 4

**Summary:**

This paper proposed some new methods to identify states from noisy observations under restrictive assumptions.

**Strengths:**

* A potentially new graphical model for POMDPs.

**Weaknesses:**

* Restrictive assumptions: The authors directly assume the model is 1-step decodable (e.g. discussed in [1]), which is indeed a very strong assumption and there are lots of existing work that can properly address it. It is not clear what's the additional benefit compared with these works. Furthermore, the assumption on the invertible emission is extremely strong that some of the previous works may not even require. I'm not convinced that such additional structural assumptions are general in practice and assuming such assumptions can be helpful.

* Unclear theoretical guarantee: It is not clear if the proposed algorithm can really identify the state from the observation. I don't think the proposed method can fully decouple state and the noise, as there will be generally no issue if we include $z_t$ into $s_t$ with a single state space model and the action do not have any impact on the noise part.

* Unclear source of benefits: Based on the previous weekness, it is unclear where the empirical benefit comes from. I want to argue that in Figure 5 it is unfair to use different observations to showcase the effectiveness. And I don't think if we use no other source of data (e.g. pair observatio data with same $s$ and different $z$ for different $o$), it is possible to exactly identify the state.

[1] Efroni, Yonathan, et al. "Provable reinforcement learning with a short-term memory." International Conference on Machine Learning. PMLR, 2022.

**Questions:**

* Is there any guarantee such that the proposed method can exactly identify the state from the observation?
* Is there any empirical evidence such that the proposed method can identify the state from the observation even with infinite number of data?

---

### Official Review · Reviewer_u1JV · 2023-10-31

**Soundness:** 1 poor
**Presentation:** 2 fair
**Contribution:** 2 fair
**Rating:** 3
**Confidence:** 4

**Summary:**

This work focuses on identifying and disentangling state and noise from observation to improve model-based RL (MBRL). This paper starts with the introduction of two propositions on the existence of state and noise estimation functions under different assumptions, where the central idea is to leverage transition preservation and reward preservation to identify state and noise. Then the propositions are used to devise a new MBRL method by proposing separate RSSM models. In the experiments, the proposed method is evaluated against DreamV3, TIA, DenoisedMDP and IFactor in DMC and RoboDest environments with noisy distractors. In addition, reconstruction quality is also demonstrated.

**Strengths:**

- The organization of this paper is satisfactory
- Several SOTA baselines are used. Reconstruction quality is demonstrated for direct evaluation.

**Weaknesses:**

The content in Section 3.1 and 3.2 is kind of verbose to me and the two propositions overlap much.
The transition from propositions to the proposed method is poor:
- The discussion right above Section 3.3 mentions how a general POMDP can be converted into a POMDP with an invertible observation function. It is good to see this but I do not see how the proposed method connects with this consideration.
- Since the transition preservation and reward preservation are approximately realized by variational reconstruction and MLE, the guarantee of optimal policy no longer exists. Especially, the practical method proposed shows a connection to the theories in DeepMDP [Gelada et al., ICML 2019].
For the proposed method, some details are missing. I did not know how $o^{s}{t}, o^{z}{t}$ are defined until I found Figure 4. I recommend the authors to move Figure 4 to the main body of paper for better clarity. Besides, how the two prior functions are selected is not clear.
The experimental evaluation and analysis are insufficient. For Table 1, three independent runs with relatively large stds make the results less convincing. Important ablations are missing, e.g., the contribution of the reward loss and the elbo loss should be ablated. I recommend the authors to add this ablation study.
Moreover, I recommend the authors to include Deep Bisimulation Control (DBC) [Zhang et al., ICLR 2021] in the experiments, as it is a representative method of learning representation for DMC with noisy distractors.



Minors:
- The first sentence of the second last paragraph of Section 3 seems unfinished.
- DreamerV3 is missing in Figure 6 while the caption mentions it.

**Questions:**

- What are the exact forms of the two prior functions used? How are the two prior functions selected?
- Can the authors add the ablation on the contribution of the reward loss and the elbo loss?

---

### Official Review · Reviewer_cMbH · 2023-11-01

**Soundness:** 2 fair
**Presentation:** 2 fair
**Contribution:** 2 fair
**Rating:** 5
**Confidence:** 3

**Summary:**

The paper draws motivation from identifying states from noise observations and then devises methodologies based on the understanding of disentangling states in partially observed MDPs. The paper provides both theory and empirical results, with an emphasis on validating the improved performance of the proposed method.

**Strengths:**

The targeted problem is interesting and of high importance. In many applications, exact state observations are impractical and partially observability is the most common modeling approach. Therefore, inferring the latent state from observations is pivotal to the policy learning in sequential decision-making problems. This work targets at separating potential noise and true latent state with a theoretical driven methodology, which is novel in my opinion.

**Weaknesses:**

Proposition 1 and 2 might need better explanations. First, we assume $\mathcal{M}$ is invertible, however, this invertibility is troublesome to me. Since $\mathcal{M}$ is a mapping that takes in state and noise and outputs an observation, how should we understand its invertibility? Is it saying that given an observation, the latent state and noise can be exactly recovered? In this case, Proposition 1 and 2 seem to be trivial, as the existence of mapping $h$ is a direct consequence of the invertibility of $\mathcal{M}$.

In addition, what is the difference in analyzing deterministic transition and stochastic transition, so that the results are presented separately? Otherwise, it looks like redundant.

The connection between the proposed RSSM and Proposition 1 and 2 is not clear to me. I have difficulty in understanding the purpose of Proposition 1 and 2, and how it relates to the improved performance in later sections.

**Questions:**

Figure 1 might miss an arrow between action and reward.

What is the expectation in Equation (1c)?

---

### Official Review · Reviewer_4Lhk · 2023-11-05

**Soundness:** 2 fair
**Presentation:** 1 poor
**Contribution:** 3 good
**Rating:** 3
**Confidence:** 5

**Summary:**

The paper deals with the problem of disentangling the part of the state that is reward-relevant from the part of the state that is reward-irrelevant (i.e., acts as noise). Prior works either do not possess guarantees, or rely on complex causal modes and strong assumptions. The current work departs from this literature by considering the observed MDP as a POMDP, where the relevant state as well as the reward-irrelevant noise are latent variables that are not observed. Assuming that there exists an invertible state/noise estimation function, so that the estimated state preserves both the transition dynamics and the reward, the authors are then able to show that the optimal policy on the derived MDP (using the estimated state but ignoring the estimated noise) will also be an optimal policy for the original MDP. Based on this result, the authors then design an algorithm by learning state and noise representations that satisfy the transition and reward preservation constraints. For this purpose, they rely on Recurrent State Space Models (RSSMs), similar to other prior works on state representation, by properly defining the transition, representation, emission and reward models. Experiments with various image observation environments, including the DeepMind Control Suite and RoboDesk, show that the proposed approach can often outperform existing algorithms, in particular in environments with a high amount of noise.

**Strengths:**

1. The modeling part of the paper is interesting and meaningful. It makes sense to rely on the 2-step dynamics of the transition and reward instead of considering long-range dependencies, since the latter can be much more complex. Furthermore, it makes sense to start from a POMDP and assume that the latent state and noise are unobserved quantities that have to be inferred. The connection that the authors make between the original POMDP and the derived MDP are not complicated but are simple to use for the purpose of state and noise disentanglement.

2. The RSSM equations are sound. The authors justify why a simpler factorized variational posterior would not work, which is not immediately obvious. That said, the novelty of the work but also the RSSM equations is not particularly high, since they mainly adapt/modify the equations from earlier works.

3. The experiments are conducted on similar environments to the prior literature, and the results look very promising. In the diverse video background scenario, the proposed method achieves large gains in 2 of the tasks (walker_run and cheetah_run), and appears to be the winner in 5 out of the 6 tasks (in finger_spin, it seems to be tied with IFactor). These gains point to the potential of the proposed framework.

4. The ablation study for symmetric vs. asymmetric structures on DMC implies that a smaller noise decoder could serve as a regularizer to ensure that the state decoder will not end up with any noise representation. This appears to be an interesting form of regularization.

5. The authors provide experiments for the noiseless setting as well as for the Robodesk environment with a rich reward-state to showcase the limitations of their work. I like the fact that the authors highlight the potential of their work, while being honest about its limitations.

**Weaknesses:**

1. It is unfortunate that there are so many language errors, which takes a negative toll on the presentation. There are a great deal of typos, incomplete sentences, missing or wrong words, or grammatical errors. It is as if the authors wrote the paper very quickly, and did not pay much attention to proofreading their manuscript. This is unfortunate, given that the work otherwise contains some interesting contributions, which are at times overshadowed by the bad presentation.

2. Propositions 1 and 2 are sound and they form a solid basis for the methodology described in the paper (i.e., the transition and reward preservation from the 1-step dynamics). That said, they are not new results. Such results have been known for quite long in the AI community, and the paper would benefit from making such connections more explicit. Some relevant papers on state abstractions:
    [1] Li, Walsh, Littman. Towards a Unified Theory of State Abstraction for MDPs.
    [2] Givan, Dean, Greig. Equivalence notions and model minimization in Markov Decision Processes, 2003.
Both of these papers discuss the fact that the optimal policy of the reduced MDP is also an optimal policy of the original (bigger) MDP. The difference is that this work starts from a POMDP but there are still significant similarities. The authors may want to cover the relevant work on state abstractions and show how their theory is connected to that prior work.

3. Objective (5) has an obvious problem: it accepts the trivial solution s_t=o_t and z_t=0. Indeed, we could trivially use the full observation as the relevant state, and assign nothing to the latent noise. In that case, the transition, representation, and emission models would be trivially true; furthermore, the reward model would also be satisfied since the reward depends on the observation. [Wang et. al., 2022] in denoised MDPs realize that, which is why they further minimize the conditional mutual information between the state and the controllable, reward-relevant part given the action sequence. Indeed, this quantity acts as a regularizer, which intuitively assigns to the relevant state only the reward-relevant components and not the  reward-irrelevant or uncontrollable ones. The current work does not mention anything about regularization. Does it apply a similar regularization? Does it rely solely on the asymmetric decoder, as stated in Strength (4) above? Such a discussion is missing from the current paper. If asymmetric decoders outperform the conditional mutual information regularizer, that would be very interesting in its own right.

4. It is obvious from the state abstraction equations that the authors define as noise the part of the observation that is both uncontrollable and reward-irrelevant (according to the typology of [Wang et. al., 2022]). In particular, z_t does not capture the part of the observation that is reward-irrelevant but controllable. This is evident in the prior p_{\psi}(z_t| z_{t-1}), which states that the evolution of z_t does not depend on the state. I believe that the authors should make this clear in their work that they target precisely the setting where z_t is both reward-irrelevant and uncontrollable, since this can clarify the scope of their work. Furthermore, some connections to prior work may be important. Z_t essentially behaves as an exogenous latent variable according to [Efroni et al., 2021] or [Dietterich, Trimponias, Chen. Reinforcement Learning with Exogenous States and Rewards, 2018.]. Essentially, the authors try to discover the exogenous latent noise, assuming it is also reward-irrelevant. Making the connections to prior work more clear would benefit this work in my opinion.

5. The authors use two distinct networks o_t^z and o_t^s and claim the reconstruct o_t from these 2 parts. It is not clear how exactly they do that. Perhaps they assume an additive decomposition, i.e., o_t=o_t^z+o_t^s? Or are o_t^z,o_t^s fend into a network that outputs o_t? The motivation is also not clear. Perhaps the authors assume that the image observation contains a noise part and then a reward-relevant part, which we can add to get the original image? Furthermore, it is not clear whether using two separate networks indeed helps performance compared to only using a single observation network that gets s_t, z_t and outputs o_t.

6. It is nice that the authors experiment on the DeepMind Control Suite. The only issue is that the experiments are quite "artificial" and not so realistic. The authors synthesize reward-irrelevance by injecting a big amount of noise, but they do not experiment with settings where the noise is much more subtle. It would be interesting to know whether the proposed algorithms could perform similarly well in typical RL tasks, where irrelevance is present but more subtle. I was wondering in particular whether in such settings the other baseline methods would perform equally well, if not better.

7. The hyperparameters are not mentioned anywhere. The authors just say that they follow the same hyperparameters as Denoised-MDP without further tuning. Mentioning the hyperparameters would be useful to make the paper self-contained. But it is also not totally clear that hyperparameters that work in one baseline should be optimal for another baseline. Spending some effort to optimize the performance of the algorithms might lead to even higher performance (for instance, in the new experimental settings, it is not obvious what the dimension of the latent noise z_t should be). I find it surprising that the authors did no hyperparameter search/optimization.

**Questions:**

1. Please proofread the manuscript carefully for typos, grammar errors, missing words, etc.

2. You could make the connection to prior work on state abstractions more explicit.

3. You could discuss more the need for regularization in the loss objective and how you achieved it in your work.

4. You could better describe the setting, i.e., you are considering setting where the noise is both uncontrollable (exogenous) as well as reward-irrelevant. You may also make stronger connections to prior work, e.g., on exogeneity.

5. You could elaborate more on the two distinct networks o_t^z and o_t^s, the reconstruction process, and the possible benefit compared to using a single network.

6. Have the authors tried less "artificial" settings? If the noise is large and diverse, other baselines have trouble learning the dynamics. But if the noise is more subtle or predictable, it is possible that this would not be the case. Perhaps some less artificial settings could shed more light on this question.

7. The authors may benefit from spending more effort on the hyperparameters, i.e., by optimizing the latent state and noise size, \alpha or \beta, etc. Mentioning the hyperparameters in the appendix would also be helpful to make the paper self-contained.